# EarthScape: A Multimodal Dataset for Surficial Geologic Mapping and Earth Surface Analysis

**Matthew A. Massey**
Kentucky Geological Survey
University of Kentucky
Lexington, KY 40506-0053
matthew.massey@uky.edu

**Nusrat Munia**
Department of Computer Science
University of Kentucky
Lexington, KY 40506-0633
nusrat.munia@uky.edu

**Abdullah-Al-Zubaer Imran**
Department of Computer Science
University of Kentucky
Lexington, KY 40506-0633
aimran@uky.edu

## Abstract

Surficial geologic (SG) maps are critical for understanding Earth surface processes, supporting infrastructure planning, and addressing challenges related to climate change and natural hazards. Advancements in artificial intelligence (AI) and the proliferation of remote sensing imagery present an opportunity to transform SG mapping and overcome many of the limitations (e.g., labor-intensive, not scalable, etc.) of current workflows. We introduce **EarthScape**, *a new AI-ready multimodal dataset designed to advance SG mapping*. EarthScape integrates digital elevation models, aerial imagery, multi-scale terrain derivatives, and vector data for hydrologic and infrastructure features. We present a complete data processing pipeline to support reproducibility and benchmarking and report baseline results across single-modality, multi-scale, and multimodal configurations. Our experiments highlight the predictive value of terrain-derived features and the challenge of generalizing across geologically diverse regions.
**Code:** https://github.com/masseygeo/earthscape
**Dataset:** https://uknowledge.uky.edu/kgs_data/16/

## 1 Introduction

Surficial geologic (SG) maps depict the spatial distribution of mostly unconsolidated materials on the Earth's surface [Compton, 1985, Lisle et al., 2011, Pavlis and Mason, 2017]. These maps are essential to address a range of contemporary challenges, such as supporting economic and national security interests in critical mineral resources [Brimhall et al., 2005, Schulz, 2017], informing mitigation and response planning for geologic hazards [Alcántara-Ayala, 2002, Van Westen et al., 2003], and providing a foundation on which to understand climate change [Anderson and Ferree, 2010]. SG maps are also relevant to more practical applications like urban land use planning [Dai et al., 2001, Hokanson et al., 2019] and engineering projects [Keaton, 2013]. Despite the demonstrable social benefit and scientific merit [Bernknopf, 1993], detailed SG maps ($\geq$ 1:100,000-scale) cover less than 14% of the United States [U.S. Geological Survey, 2025].

The modern SG mapping workflow relies on manual fieldwork and visual interpretation of remote sensing (RS) imagery [Compton, 1985, Lisle et al., 2011]. Because these maps rely on visual interpretation and field annotation, they often reflect expert judgment rather than reproducible criteria,

Submitted to 39th Conference on Neural Information Processing Systems (NeurIPS 2025). Do not distribute.

complicating efforts to scale mapping to national or global extents [Jones et al., 2004]. Finally, financial resources prevent large-scale initiatives to collect and compile SG map data, where one standard 1:24k-scale map may cost up to $123k [Berg, 2025]. In Kentucky alone, despite prioritizing SG mapping since 2004, fully mapping the remainder of the state at the current pace and workforce capacity would require over 175 years and an estimated $31 million [U.S. Geological Survey, 2024b].

Advancements in deep learning and the proliferation of RS imagery present an opportunity to transform SG mapping, overcoming current limitations. Recent studies have showcased the power of this type of approach to identify or segment landslides [Prakash et al., 2021, Wang et al., 2021, Liu et al., 2023] and sinkholes [Rafique et al., 2022]. Several studies have extended these ideas to segment maps of multiple classes of geologic materials [Behrens et al., 2018, Latifovic et al., 2018, Wang et al., 2021, Liu et al., 2024b]. These studies have demonstrated the utility of computer vision (CV) for geological investigations, but this area of research is still in its infancy.

The challenges presented by SG mapping align closely with current trends in CV research. Multi-modal fusion of diverse geological datasets is necessary to accurately capture geologic map features [Baltrušaitis et al., 2018, Steyaert et al., 2023, Li and Wu, 2024]. The spatial dependencies of geological features resonate with recent advances in attention mechanisms [Dosovitskiy, 2020, Niu et al., 2021, Hassanin et al., 2024], multi-scale architectures [Chen et al., 2017, Fan et al., 2021, Liu et al., 2024a], and contrastive learning frameworks [Chen et al., 2020, Le-Khac et al., 2020, Song et al., 2024] that capture context and structural relationships. Moreover, the scale-dependent and highly localized nature of geological processes demands robust methods for handling extreme class imbalance and ensuring geographic generalizability [Ghosh et al., 2024, Lin, 2017].

The rapid progress in CV has been driven primarily due to the availability of large-scale, standardized datasets. General-purpose benchmarks, such as ImageNet [Deng et al., 2009] and COCO [Lin et al., 2014], have catalyzed advances in classification, detection, and segmentation by offering vast repositories of labeled imagery and clear evaluation protocols. However, performance on real-world, domain-specific tasks often plateaus without datasets that reflect their unique characteristics, sensing modalities, and physical constraints. In the geospatial domain, several specialized datasets have emerged for land cover classification and urban scene analysis [Schmitt et al., 2019, Cordts et al., 2016, Demir et al., 2018, Van Etten et al., 2018, Sumbul et al., 2019]. But they are primarily focused on detecting anthropogenic features and land use. Only a single publicly available geologic dataset exists, and it is limited to landslide detection from a narrow set of features [Ji et al., 2020]. This underscores a critical gap in datasets tailored for Earth surface processes.

EarthScape is a multimodal dataset developed for SG mapping, with broad applicability to planetary surface analysis. It integrates publicly available overhead RGB and near-infrared (NIR) imagery, digital elevation models (DEMs), geomorphometric terrain features derived at multiple spatial scales, and transportation and hydrological networks from vector geographic information system (GIS) sources. *This multimodal, multi-scale design captures the complexity of Earth surface (ES) processes and provides a robust benchmark for advancing multimodal learning, geospatial vision, and geological analysis.* Our specific contributions are summarized as follows:

- EarthScape, the first AI-ready dataset specifically designed for SG mapping and ES analysis.

- Design and release of a rich set of input features that span multiple spatial scales and modalities, enabling models to learn representations of surface shape that generalize better across local and regional terrain variations.

- Establishing baseline benchmarks for multilabel classification using both unimodal and multimodal configurations. These include individual modality tests, multi-scale fusion within a single modality, and cross-modality fusion strategies.

## 2 Related work

**SG Mapping with Machine Learning:** SG mapping focuses on unconsolidated materials formed by active surface processes such as weathering, erosion, sediment transport, and deposition [Compton, 1985, Lisle et al., 2011, Pavlis and Mason, 2017]. These materials are closely tied to landform structure and surface morphology, as terrain shape governs the energy available to drive these processes and influences the way sediments are generated, transported, and deposited [Odeh et al., 1991, Schomberg

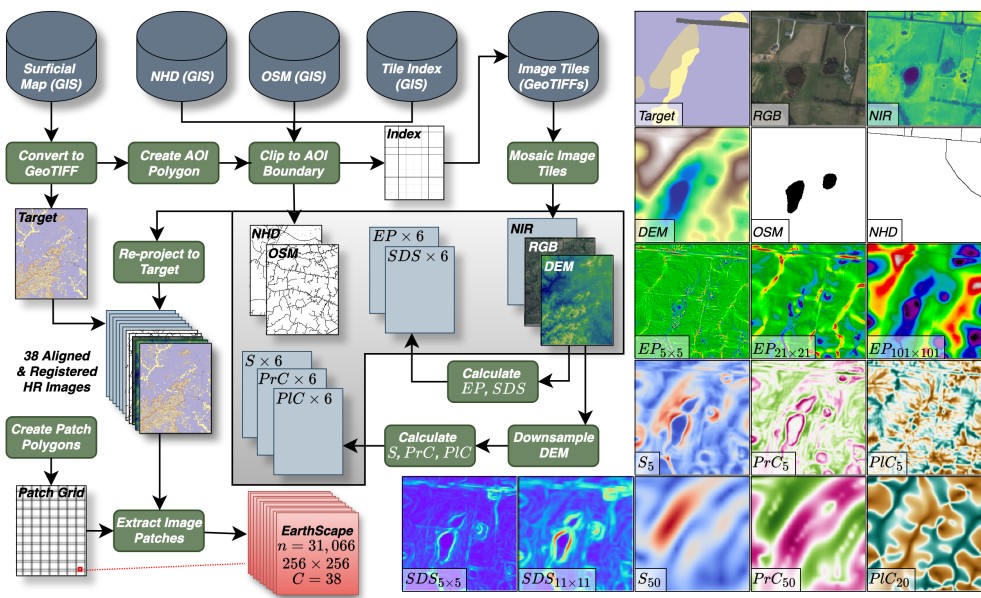

Figure 1: EarthScape data processing pipeline (left) and selected modalities from a single $256 \times 256$ patch (right). The SG map target is rasterized and used to define the area of interest (AOI), from which all predictive features (DEM, RGB+NIR imagery, NHD hydrology, and OSM infrastructure) are clipped and aligned. Terrain derivatives are then computed from the DEM at multiple spatial scales. A regular grid is applied to extract 38 co-registered channels per patch.

et al., 2005, Brigham and Crider, 2022]. Several studies have leveraged this terrain-material relationship using traditional machine learning methods, including logistic regression, random forests, and support vector machines, for classification or pixel-wise segmentation of single-class features (e.g., landslides, sinkholes) [Kirkwood et al., 2016, Zhu and Pierskalla Jr, 2016, Crawford et al., 2021] or multiclass geologic maps [Cracknell and Reading, 2014, Johnson and Haneberg, 2025]. However, these models rely on hand-crafted features, are limited to small geographic extents, and fail to generalize beyond the training region.

More recently, deep learning approaches using convolutional neural networks (CNNs) and CNN-transformer hybrids have been applied to these tasks [Prakash et al., 2021, Ji et al., 2020, Liu et al., 2023, Latifovic et al., 2018, Zhou et al., 2023, Rafique et al., 2022]. While these models better capture spatial dependencies critical to geologic interpretation [Bishop et al., 1998, Behrens et al., 2018], they remain constrained to narrow geographic domains, lack publicly available datasets or reproducible pipelines, and often rely on limited input modalities.

**Remote Sensing Datasets:** RS benchmarks such as SpaceNet [Van Etten et al., 2018], xView [Lam et al., 2018], and Functional Map of the World [Christie et al., 2018] provide high-resolution satellite imagery annotated for object detection and scene classification in urban environments. These datasets are optimized for anthropogenic features such as roads, buildings, and vehicles, and are widely used for infrastructure monitoring and disaster response. Other RS datasets like BigEarthNet [Sumbul et al., 2019], DeepGlobe [Demir et al., 2018], and SEN12MS [Schmitt et al., 2019], extend the domain to land cover classification and segmentation using multispectral or synthetic aperture radar (SAR) imagery. However, these datasets target coarse semantic categories like vegetation or developed areas, rather than physical topographic characteristics. These datasets lack representations of Earth's surface, which are essential for interpreting geological processes.

**Multimodal Learning for Geologic Tasks:** Multimodal learning has become a central paradigm in RS and geospatial CV, where combining diverse data sources like optical imagery, SAR, and DEMs can enhance model robustness through complementary information [Astruc et al., 2024, Bi et al., 2022, Jain et al., 2022, Han et al., 2024]. In geological applications, this often involves pairing overhead RGB imagery with DEMs, fused using early- or mid-level strategies [Prakash et al., 2021, Ji et al., 2020, Liu et al., 2023, Latifovic et al., 2018, Zhou et al., 2023, Rafique et al., 2022]. These

modalities have proven effective for detecting recent geomorphic events such as landslides, where strong topographic and visual signals are present. However, model performance often deteriorates when features are older, vegetated, or eroded, limiting their interpretability and transferability [Ji et al., 2020, Liu et al., 2023, Zhou et al., 2023].

Several studies have explored additional modalities such as elevation contours [Zhou et al., 2023], geochemical field data [Latifovic et al., 2018, Wang et al., 2021], and aeromagnetic imagery [Liu et al., 2024b]. While successful, these studies were site-specific, and the datasets are not commonly available or standardized for machine learning workflows. Rafique et al. [2022] evaluated several elevation-based parameters, including DEMs, slope, and shaded relief, finding that raw DEMs performed best. However, their geographically-limited validation suggests that models may have relied on absolute elevation rather than generalizable terrain patterns.

## 3 EarthScape Dataset

### 3.1 Composition and Features

**Surficial Geologic Maps:** The Kentucky Geological Survey has conducted high-resolution SG mapping since 2004, targeting rapidly developing regions and transportation corridors across the state. Mapping is performed at a scale of 1:24,000 or finer, widely considered the gold standard for detailed geological surveys. The EarthScape dataset currently includes SG map data from Warren and Hardin Counties [Buchanan et al., 2023, Massey et al., 2023, Swallom et al., 2023, Massey et al., 2024, Hodelka et al., 2024, Swallom et al., 2024, Bottoms et al., 2021, Massey et al., 2021], which provide the multilabel targets and segmentation masks (Fig. 1). Seven SG map units are represented, capturing three dominant surface processes: fluvial deposition, gravitational transport, and in-situ weathering. These include _alluvium (Qal)_ and _terrace deposits (Qat)_ from river activity; _alluvial fans (Qaf)_ associated with debris flow hazards; _colluvium (Qc)_ and _colluvial aprons (Qca)_ from hillslope processes; _residuum (Qr)_ from bedrock weathering; and _artificial fill (af1)_ from anthropogenic modification. All maps are publicly available as vector polygons in ESRI file geodatabase format. See the supplement for detailed unit descriptions.

**Aerial imagery and DEM:** The KyFromAbove program has been acquiring high-resolution aerial imagery and DEMs for the state of Kentucky, USA since 2010 [Commonwealth of Kentucky, 2024]. Aerial imagery consists of RGB and NIR channels with a 6-inch spatial resolution. Its utility is in identifying anthropogenic features (such as af1) that are easily distinguished from natural landscapes (Fig. 1). The NIR band further enhances the detection of hydrological features, such as alluvial deposits and stream channels, by highlighting vegetation patterns that can indicate water presence or recent sediment deposition (Fig. 1). However, the utility of aerial RGB and NIR in delineating detailed SG map units is limited. In contrast, the DEM, generated from airborne LiDAR with a 5ft/pixel spatial resolution, is a critical feature for SG mapping and ES analysis (Fig. 1). Both the DEM and the aerial imagery are available as publicly accessible GeoTIFF tiles.

**Geomorphometric Terrain Features:** The DEMs provide a foundation for deriving five key terrain features widely used in geomorphometric analysis and essential for delineating SG units (Fig. 1) [Florinsky, 2016]. These include: _slope (S)_ measures terrain steepness; _profile curvature (PrC)_ and _planform curvature (PlC)_ are directional second derivatives capturing flow acceleration and divergence; _elevation percentile (EP)_ is a relative topographic position metric; _standard deviation of slope (SDS)_ is a measure of terrain roughness quantifying local variability of slope angles. Each feature was calculated at multiple spatial scales to capture both localized and regional landform structure (see supplement for detailed definitions and scale parameters).

**Hydrography and Infrastructure:** To support downstream tasks involving fluvial and anthropogenic processes, EarthScape includes vector data for hydrographic and infrastructure features (Fig. 1). Stream centerlines and waterbody polygons from the U.S. Geological Survey's National Hydrography Dataset (NHD) [U.S. Geological Survey, 2024a] provide context for identifying alluvial units within stream valleys. Road and railway centerlines from OpenStreetMap (OSM) [OpenStreetMap contributors, 2024] delineate areas modified by human activity, such as artificial fill. These features also help characterize geologic disturbance near infrastructure, including slope undercutting and landslide susceptibility. Both datasets are included as binary raster channels aligned to the patch grid.

## 3.2 Data Processing

**Targets:** Each SG map was downloaded as a vector GIS geodatabase, the relevant feature class extracted, and the vector polygons inspected for topological correctness, ensuring no overlaps, no gaps, and valid polygon geometries (Fig. 1). The validated data was saved as a standalone GeoJSON file, which was then used to generate a boundary polygon defining the area of interest (AOI) for clipping and extracting relevant portions of other datasets. SG target classes were encoded with ordinal values in the GeoJSON. Finally, the vector GeoJSON was rasterized to a GeoTIFF image with a 5ft/pixel spatial resolution, matching the native resolution of the DEM.

**Features:** Vector datasets, including NHD, OSM, and the KyFromAbove tile index, were downloaded, clipped to the target AOI, and saved as standalone GeoJSON files (Fig. 1). NHD stream centerlines and waterbody polygons, and OSM road and railway centerlines were rasterized into two binary GeoTIFFs representing hydrography and infrastructure, respectively. The KyFromAbove tile index defines the locations of aerial RGB, NIR, and DEM data tiles across Kentucky. Using the AOI, relevant locations were selected and the corresponding image tiles downloaded (Fig. 1). DEM tiles were merged into a single GeoTIFF mosaic at 5ft/pixel resolution. RGB and NIR imagery underwent similar processing, with additional downsampling from 6in/pixel to 5ft/pixel resolution.

Five terrain features were calculated at six different spatial scales directly from the DEM mosaic (Fig. 1). $S$, $PrC$, and $PlC$ were created using $5 \times 5$ kernels applied to the original 5ft/pixel DEM and five additional DEMs downsampled with cubic convolution to resolutions of 10, 20, 50, 100, and 200ft/pixel. A Gaussian filter was applied to each downsampled DEM to smooth potential artifacts, the relevant terrain feature was calculated, then upsampled back to the original resolution of 5ft/pixel using cubic convolution, and another Gaussian filter was applied to minimize resampling artifacts. $SDS$ and $EP$ were calculated using six kernel sizes of $5 \times 5$, $11 \times 11$, $21 \times 21$, $51 \times 51$, $101 \times 101$, and $201 \times 201$ pixels, applied only to the original 5ft/pixel DEM. These kernel sizes capture receptive fields similar to those represented by the coarser-resolution DEMs used for $S$, $PrC$, and $PlC$, but are better suited for $SDS$ and $EP$ due to their reliance on the number of neighbors.

**Spatial Alignment and Registration:** The target SG map GeoTIFF images served as the spatial reference for aligning all other features in the dataset (Fig. 1). Once each feature was collected and compiled into its respective GeoTIFF image file, they were reprojected to align with the reference image coordinates using cubic convolution interpolation. All images were checked to ensure that their bounding coordinates and spatial resolutions were identical across all other images.

**Image Patches:** Vector polygon patches were systematically constructed in a grid pattern to cover the target AOI using the same coordinate reference system as the target GeoTIFF (Fig. 1). Each grid cell polygon was assigned a unique patch ID, and then all patches were saved as a GeoJSON file. Each grid cell polygon patch was constructed so that it covers an area of exactly $1280 \times 1280$ feet ($256 \times 256$ pixels), overlaps adjacent patches by 50%, and is fully contained within the target AOI. Each cell was assigned a unique patch ID and used to extract 38 corresponding channels, including target mask, aerial RGB and NIR, DEM, the five terrain features calculated at six scales, NHD, and OSM. Target masks were then used to extract one-hot encoded class labels and the proportional areas occupied by each class within each patch.

## 3.3 Dataset and Statistics

EarthScape currently comprises 31,018 image patch locations, each measuring $256 \times 256$ pixels with 50% spatial overlap with adjacent locations (Fig. 1). Each patch contains 38 channels, stored as individual 32-bit float GeoTIFF files with embedded geospatial metadata. Patch geometries are defined in an accompanying GeoJSON file to support spatial querying and GIS-based evaluation. The dataset spans two regions in Kentucky: a large contiguous subset of 23,566 locations in Warren County (Fig. 2A) and 7,452 locations in Hardin County. This geographic partitioning enables cross-region generalization studies and domain adaptation experiments, with additional regions planned as new SG maps become available. The dataset exhibits significant spatial and statistical heterogeneity. Most patches contain multiple SG units, with up to six unique classes per patch, and pronounced spatial variability across the AOIs in class co-occurrence (Fig. 2A, 2D). The dataset is highly imbalanced, with common units like Qr dominating the distribution and minority classes Qaf and Qat appearing infrequently (Fig. 2B). Intra-patch complexity is further reflected in the proportional area each class occupies per patch (Fig. 2C), with many units contributing small but

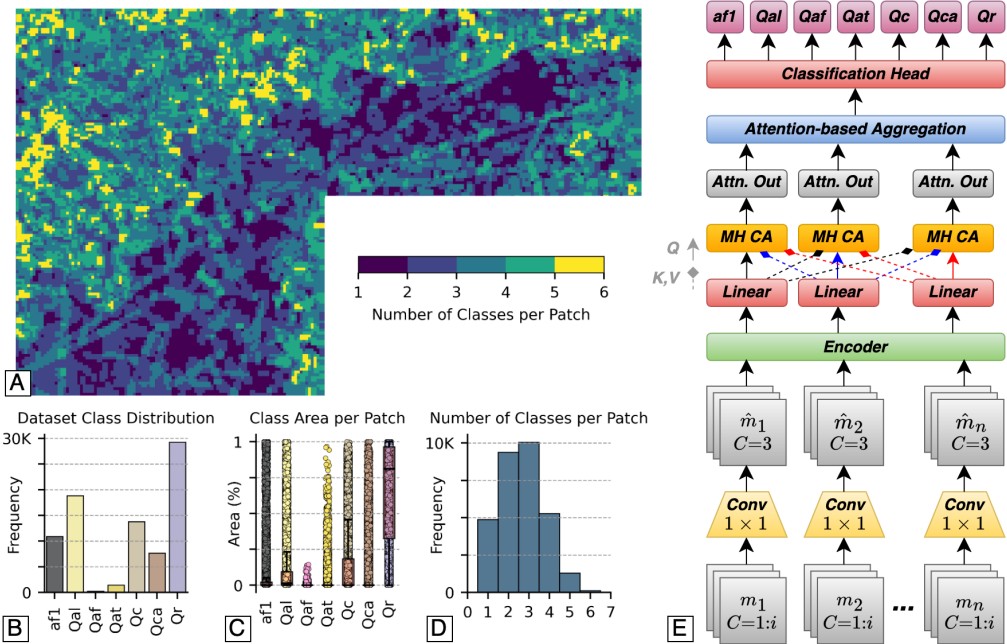

Figure 2: EarthScape dataset characteristics (A–D) and SGMap-Net architecture (E). A. Choropleth map of Warren County showing the number of classes per patch, illustrating spatial heterogeneity and multilabel complexity. B. Dataset-wide class distribution, highlighting significant class imbalance. C. Proportional area of each class per patch, showing that many patches include low-exposure classes, increasing classification difficulty. D. Histogram of class counts per patch, further illustrating multilabel and intra-patch complexity. E. SGMap-Net architecture comprising a standardization module, shared encoder, and multilabel classification head. Fusion is implemented via early channel stacking and intermediate attention-based strategies.

meaningful fractions to the total label. These properties make EarthScape well-suited for evaluating multilabel models under realistic geological class imbalance and spatial heterogeneity.

## 4 Experiments

### 4.1 Methods

**Task Definition:** We formulate SG mapping as a multilabel classification task over multimodal geospatial inputs. Each input sample corresponds to a $256 \times 256$ image patch with co-registered modalities and a label vector indicating the presence or absence of each of the SG units. Let $\mathcal{D} = (x_i, y_i)_{i=1}^{N}$ denote the dataset, where each $x_i = m_1, m_2, \ldots, m_n$ is a collection of $n$ modality-specific input tensors (e.g., DEM, $EP$, $PlC$, etc.) and each modality $m_i$ can have multiple scaled images that we consider as channels $C_i$. The $y_i \in 0, 1^K$ is a binary label vector over $K = 7$ classes. The model learns a mapping $f : X \to [0, 1]^K$ to predict per-class probabilities, enabling multi-class label assignment for each patch. This formulation allows us to systematically evaluate how different modality combinations contribute to geologic feature recognition and serves as a tractable benchmark for future tasks such as semantic segmentation.

**Surficial Geologic Mapping Network (SGMap-Net):** Our dataset comprises multiple geospatial image modalities with varying channel dimensionalities (e.g., RGB, DEM, terrain derivatives), which we aim to classify into seven geologic classes. To effectively integrate the complementary information across modalities, we propose a fusion-based model, SGMap-Net, which incorporates both early and intermediate fusion mechanisms to capture fine-grained spatial cues and high-level semantic relationships. Figure 2 (E) illustrates the overall architecture of SGMap-Net, which consists of three key components: a standardization module, a feature extractor, and a classification head. As part of our early fusion strategy, we first stack all channels of each modality $m_i$ and then apply a

$1 \times 1$ convolution followed by batch normalization and ReLU activation to standardize the input to a common channel dimension $C = 3$. This ensures compatibility with a shared encoder while preserving modality-specific spatial patterns through independent convolutions.

$$\hat{m}_i = ReLU(BN(Conv1 \times 1(m_i))). \tag{1}$$

Each standardized modality $\hat{m}_i$ is passed through a shared encoder to extract feature maps $f_{m_i} = Encoder(\hat{m}_i)$. The shared encoder is initialized with ImageNet-pretrained weights, and we experiment with ResNeXt-50 [Xie et al., 2017] and Vision Transformer (ViT-B/16) [Dosovitskiy, 2020] architectures. Next, each feature vector $f_{m_i}$ is projected into a common latent space of dimension $d$ using a fully connected layer and augmented with a learnable modality embedding $e_i$ to get the final representaions $z_i = f_{m_i} + e_i$. Then we apply modality-specific multi-head attention (MHA) [Vaswani et al., 2017] mechanisms to enable intermediate fusion across modalities. For each modality $m_i$, attention is computed using $z_i$ as the query $(Q)$, and the embeddings from all other modalities as keys $(K)$ and values $(V)$.

$$a_i = MHA(Q = z_i, K = [z_j]_{j \neq i}, V = [z_j]_{j \neq i}). \tag{2}$$

Next, we perform attention-weighted aggregation over the set of modality-specific attention outputs $a$. We begin by concatenating all outputs $A = [a_i]$. To determine the relative importance of each modality, we apply a learnable linear projection $v_i$ followed by a softmax operation to obtain attention weights $w = Softmax(v^T A)$. The final fused representation is then computed using these weights, $z_{fused} = \sum_{i=1}^{N} w_i a_i$. This attention-weighted aggregation adaptively emphasizes the most informative modalities for each sample. The fused embedding $z_{\text{fused}}$ is then passed through a classification head consisting of two fully connected layers to predict the geologic class logits $\hat{y}$. In addition to our proposed attention-based fusion strategy, we evaluate two alternative approaches, cross-modality channel stacking and concatenation. We stack selected channels from different modalities, extract a joint representation using the encoder, and feed it into the classification head. In another approach, we concatenate the modality embeddings from the encoder and pass them directly to the classification head. These variants serve as comparative baselines to assess the impact of modality-aware attention in our fusion framework.

**Data Splits and Selection:** We define training, validation, and in-domain test splits using the Warren County subset. A total of 1,536 patch locations were randomly selected for the in-domain test set. Next, 768 non-intersecting locations were randomly sampled for validation. All remaining patches that did not intersect the in-domain test patches or validation patches were used for training (8,416). To evaluate geographic generalization to a geologically similar, but previously unseen region, we sampled an additional cross-domain test set of 1,536 patches from the Hardin County subset. While this split uses less than half of the available EarthScape patches, it was chosen to balance typical dataset proportions and maintain spatial independence between training and evaluation regions.

**Training Procedure:** All patches were normalized using modality-specific means and standard deviations computed over the in-domain dataset to ensure consistent input scaling. Data augmentation included random horizontal and vertical flips and 90◦ rotations, reflecting that geologic features are not orientation-dependent. Restricting rotations to right angles preserves label accuracy by preventing small classes along edges from being cropped due to padding. To address class imbalance, we adopted focal loss [Lin, 2017] with $\alpha = 0.25$ and $\gamma = 2.0$ for all experiments. Oversampling was tested but degraded performance, so training used the original distribution. Models were trained for 15 epochs using the Adam optimizer, a fixed learning rate of 0.001, and batch size of 16. The model with the lowest validation loss was used for testing. After training, label-wise thresholds were optimized for F1 on the validation set and applied to both in-domain (Warren) and cross-domain (Hardin) test sets. Performance was evaluated using per-class and macro-averaged accuracy, precision, recall, F1 score, average precision (AP), and area under the ROC curve (AUC). See the supplemental material for focal loss tuning, training time, and compute details.

## 4.2 Results and Discussion

**Single Modality Benchmarks:** We first evaluated single-modality models using SGMap-Net with both ResNeXt-50 and ViT-B/16 backbones (Table 1; also see supplemental material). Among the ResNeXt-50 models, the best in-domain performance was achieved using $EP\ 51 \times 51$, $EP\ 5 \times 5$, and $EP\ 21 \times 21$, all of which outperformed the top ViT models. Most classes benefited from the relative elevation signal captured by $EP$, except for Qc, which performed best with slope $(S)$,

Table 1: Macro-averaged F1 scores, precision, AUC, and accuracy on in-domain (Warren County, WC) and cross-domain (Hardin County, HC) test sets, along with differences between WC and HC ($\Delta$) for each metric. Results are reported for the top three models in each experimental setting: single-modality, multi-scale fusion, and multimodal. Parentheses indicate the spatial scales used for fusion—ms: all spatial scales; s: smallest only; l: largest only. All models use a ResNeXt backbone and fusion with early channel stacking. The best and second-best scores in each column are indicated in **bold** and underlined, respectively. Additional results are provided in the supplemental material.

| Model | F1 | | | Precision | | | AUC | | | Accuracy | | |
|---|---|---|---|---|---|---|---|---|---|---|---|---|
| | WC | HC | $\Delta$ | WC | HC | $\Delta$ | WC | HC | $\Delta$ | WC | HC | $\Delta$ |
| $EP\ 51 \times 51$ | 0.651 | 0.380 | 0.271 | 0.612 | 0.382 | 0.230 | 0.876 | 0.663 | 0.213 | 0.862 | 0.818 | 0.044 |
| $EP\ 5 \times 5$ | 0.648 | 0.357 | 0.291 | 0.617 | 0.450 | 0.167 | 0.872 | 0.582 | 0.290 | 0.858 | 0.831 | 0.027 |
| $EP\ 21 \times 21$ | 0.645 | 0.384 | 0.261 | **0.629** | 0.455 | 0.174 | 0.877 | 0.695 | 0.182 | 0.860 | 0.828 | 0.032 |
| $EP\ (ms)$ | 0.640 | 0.425 | 0.215 | 0.606 | **0.556** | **0.050** | 0.862 | 0.717 | 0.145 | 0.865 | 0.828 | 0.037 |
| $S\ (ms)$ | 0.637 | 0.594 | **0.043** | 0.607 | 0.535 | 0.072 | 0.864 | 0.804 | 0.060 | 0.856 | 0.860 | **-0.004** |
| $SDS\ (ms)$ | 0.636 | 0.588 | 0.048 | 0.588 | 0.509 | 0.079 | 0.878 | 0.792 | 0.086 | 0.846 | 0.839 | 0.007 |
| $EP+S+SDS\ (ms)$ | **0.657** | **0.598** | 0.059 | 0.626 | 0.546 | 0.080 | 0.882 | 0.806 | 0.076 | **0.875** | **0.867** | 0.008 |
| $EP+S+SDS\ (s)$ | 0.641 | 0.568 | 0.073 | 0.606 | 0.531 | 0.075 | 0.848 | **0.812** | **0.036** | 0.865 | 0.856 | 0.009 |
| $EP+S+SDS\ (l)$ | 0.626 | 0.582 | 0.044 | 0.588 | 0.529 | 0.059 | **0.885** | **0.812** | 0.073 | 0.858 | 0.852 | 0.006 |

aligning with its gravity-driven depositional process. Rare classes such as Qaf and Qat remained poorly identified across all experiments, suggesting the need for targeted loss strategies, additional training data, or synthetic augmentation Across all modalities, DEM-derived features $EP$, $S$, and $SDS$ consistently outperformed raw DEM inputs, reinforcing the value of domain-specific terrain derivatives over implicit feature learning. Additionally, no single spatial kernel was optimal across all classes (e.g., af1 performed best with $EP\ 11 \times 11$, while Qal favored $EP\ 51 \times 51$), highlighting the importance of multi-scale inputs. Model performance declined under cross-domain testing in Hardin County. However, the ViT backbone showed better generalization ($\Delta F1_{ViT} = 0.018$ vs. $\Delta F1_{ResNeXt} = 0.043$). $S$ and $SDS$ exhibited the best cross-region transfer, while raw DEM inputs underperformed, likely due to overfitting region-specific topography.

**Multi-scale Fusion:** Unimodal experiments showed that no single spatial scale consistently performed best across all classes, with each SG map unit exhibiting distinct preferences for both modality and resolution. To explore whether combining spatial scales could improve performance, we evaluated the effects of fusing all six spatial resolutions for terrain features. Models were trained using both ResNeXt-50 and ViT-B/16 backbones (Table 1; also see supplemental material). Early fusion with channel stacking and ResNeXt yielded the most reliable results. For $SDS$, fusion slightly improved in-domain F1 (from 0.633 to 0.636) and enhanced cross-domain generalization ($\Delta F1$ decreased from 0.060 to 0.048). $EP$ experienced a modest drop in in-domain F1 (0.651 to 0.640), but showed a substantial improvement in generalization ($\Delta F1$ decreased from 0.271 to 0.215), indicating that multi-scale fusion can mitigate its sensitivity to regional relief variation. Mid-level attention-based fusion underperformed in all cases, suggesting that early fusion is both more effective and more stable for combining spatial scales.

**Multimodal Fusion:** We evaluated multimodal fusion using both ResNeXt-50 and ViT-B/16 backbones, testing three fusion strategies: early fusion via channel stacking, mid-level attention-based fusion, and mid-level fusion via feature concatenation. We tested three modality configurations: (1) RGB + DEM, a common baseline in geospatial literature; (2) $EP + S + SDS$, selected based on unimodal performance; and (3) a full configuration combining DEM, RGB, $EP$, $S$, and $SDS$. For the $EP + S + SDS$ configuration, we tested three variants: one using all six spatial scales for each modality, one with three representative scales, and one with a single scale per modality (Table 1; also see supplemental material).

The best-performing model, according to both in-domain and cross-domain F1, was the $EP + S + SDS$ configuration using three selected spatial scales. While the in-domain macro-F1 improved only modestly (from 0.651 to 0.657), the cross-domain F1 increased dramatically from 0.380 to 0.598, reducing the generalization gap ($\Delta F1$) from 0.271 to just 0.059. This result underscores the strength of terrain-based, multi-scale inputs for learning region-invariant surface structure. Two reduced variants of the same modality set, using single scales per modality, ranked second and third, confirming the robustness of shape-centric features and the benefit of multi-scale representations. The next best performers were the $EP + S + SDS$ models using mid-level concatenation, followed by the full model (DEM+RGB+$S + SDS + EP$) with the same fusion strategy. In contrast, the

RGB+DEM configuration performed worst across all fusion methods and backbone combinations, reinforcing the limited generalizability of location-sensitive visual and elevation inputs. Despite its architectural sophistication, the attention-based fusion strategy consistently underperformed, suggesting that early fusion, and even simpler mid-level concatenation, can be more effective than complex attention mechanisms for integrating geospatial modalities in this domain.

## 5 Challenges and Limitations

**Geographic Scope and Extensibility:** EarthScape is currently limited to two regions in Kentucky, USA, reflecting both the availability of high-resolution SG maps and the natural variability of geological processes. While this constraint is typical of geospatial datasets, EarthScape was designed with a modular, patch-based architecture to support expansion. Kentucky is the only state in the region with SG maps of this terrain type available in standardized GIS formats, but the dataset curation workflow is broadly applicable. Ongoing efforts aim to incorporate additional regions and globally available features to improve geographic coverage and enable cross-domain model development.

**Class Imbalance:** The dataset includes seven SG units with highly imbalanced distributions that reflect real-world conditions. At the patch level, the number of co-occurring classes ranges from one to six, and many units occupy only a small fraction of a given patch. This results in both inter-class imbalance and intra-patch heterogeneity, offering a challenging testbed for multilabel and segmentation models that must handle sparse and noisy labels.

**Geographic Generalization:** SG varies significantly across regions due to localized geomorphic and depositional processes. Unlike many AI benchmarks that assume spatial homogeneity, EarthScape explicitly supports the evaluation of cross-region generalization. The inclusion of two distinct geographic subsets allows for benchmarking spatial transfer and domain adaptation performance under realistic conditions.

**Multi-scale Complexity:** SG features are scale-dependent, with different processes operating at distinct spatial resolutions. EarthScape includes terrain derivatives computed at six spatial scales, enabling models to learn both local and regional landform patterns. This supports research in multi-scale fusion, resolution-aware architectures, and feature relevance across spatial hierarchies.

**Interpretation Variability:** Although EarthScape relies on expert-labeled SG maps, geological interpretation is inherently uncertain, particularly in regions with limited field validation or ambiguous unit boundaries. This introduces structured label noise, which poses a challenge for supervised learning but also provides an opportunity to develop models that are robust to real-world uncertainty.

**Temporal Inconsistency:** The DEM, imagery, and vector layers were acquired between 2019 and 2024, introducing potential temporal mismatches across modalities. While this may reduce fine-grained alignment in some patches, it offers an opportunity to evaluate model resilience to asynchronous data and supports future work in temporal generalization.

## 6 Conclusions

We introduced EarthScape, a new AI-ready, multimodal benchmark dataset for SG mapping and ES analysis. EarthScape integrates aerial imagery, DEMs, multi-scale terrain derivatives, and GIS vector data, offering a unique resource for multimodal geospatial learning. The dataset presents real-world challenges like class imbalance, spatial heterogeneity, and geographic variability, making it a robust testbed for developing and evaluating AI models. Through baseline experiments, we established performance benchmarks across individual modalities, multi-scale fusion, and multimodal inputs, highlighting both the predictive value of terrain-based features and the difficulty of cross-region generalization in geologic settings. Designed as a living dataset, EarthScape is extensible in both geographic and modality space. Ongoing work includes expanding regional coverage, incorporating globally available features, and improving fusion strategies. Future directions include high-resolution segmentation tasks, pretraining pipelines, and region-specific fine-tuning to support applied geological workflows. By releasing data, code, and benchmarks, we aim to foster reproducible research, cross-disciplinary collaboration, and the development of generalizable models for geospatial AI.

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
