# OpenReview forum: "EarthScape: A Multimodal Dataset for Surficial Geologic Mapping and Earth Surface Analysis"
_NeurIPS.cc/2025/Datasets_and_Benchmarks_Track — Submitted to NeurIPS 2025 Datasets and Benchmarks Track_

### Official Review · Reviewer_3sWo · 2025-06-29

**Rating:** 3
**Confidence:** 3

**Summary:**

This paper presents EarthScape, a multimodal dataset and a baseline model (SGMap-Net) for surficial geologic (SG) mapping using remote sensing imagery and derived terrain features. While the paper targets an important scientific application with societal impact, there are several limitations in the methodological contributions and experimental rigor that should be addressed.

**Dataset Code Accessibility:**

Yes

**Ethical Considerations:**

No, there are no or only very minor ethics concerns

**Final Justification:**

As mentioned in the discussion, the dataset in the paper is not particularly distinctive in terms of scope, scale, or task. Therefore, I am inclined to recommend rejection.

**Limitations Weaknesses:**

1. Lack of Methodological Innovation:
The proposed SGMap-Net model lacks clear design motivation or theoretical justification. It essentially combines early fusion (via channel stacking) and intermediate fusion (via cross-modality attention), but these are standard techniques in multimodal literature. There is little evidence that SGMap-Net is tailored to the specific challenges of SG mapping beyond superficial architectural choices. (But this might be an overly strict expectation for the DB track.)

2. Baseline Comparison is Incomplete:
The experimental comparison is primarily limited to ResNeXt-50 and ViT-B/16 backbones within the proposed framework. There is no comparison against other state-of-the-art multimodal fusion models or even competitive standalone methods (e.g., UNet, DeepLab, or recent transformer-based segmentation models). This severely limits the ability to assess the effectiveness of SGMap-Net.

3. Dataset Contribution is Overstated for the ML Community:
While the dataset is carefully curated and domain-relevant, from an ML standpoint, it reduces to a moderately sized patch-level multilabel classification dataset with seven classes and 31k samples. This scale and task formulation do not present a substantial technical challenge compared to existing vision datasets. Its novelty is mostly in the domain (geology), not the ML methodology.

4. Limited Geographic Scope and Generalizability:
Despite the authors’ efforts to include two regions for domain generalization, the dataset remains geographically narrow (both counties are in Kentucky). The paper acknowledges this limitation but does not convincingly argue how this affects ML model robustness beyond the initial use case.

**Strengths Contributions:**

The EarthScape dataset is well-constructed, integrating DEMs, aerial imagery, terrain derivatives, and infrastructure features. It is clearly valuable for the geology and remote sensing communities.

The authors provide a detailed data processing pipeline and ensure reproducibility through public release of data and code.

The paper carefully discusses challenges such as geographic generalization, class imbalance, and multi-scale fusion, which are indeed realistic in geoscience tasks.

---

> ### Author Rebuttal · Authors · 2025-07-31
>
> We are grateful for the detailed and constructive feedback from Reviewer 3sWo, and for recognizing the diversity of EarthScape's (ES) geospatial features and the breadth of our multimodal and multi-scale experiments.
>
> # "1. Lack of Methodological Innovation"
>
> Our introduced ES dataset offers the use of topological features for surficial geologic mapping that is currently not possible with the existing geologic datasets. Unlike the existing multimodal models, SGMap-Net is explicitly designed to integrate these modalities for geologic context, capturing critical shape-based information in surficial geologic mapping. The chosen fusion strategies in the proposed SGMap-Net are directly motivated by the unique characteristics of the modalities in our dataset, e.g., resolution, modality sparsity, etc. Moreover, our comprehensive evaluation across in-domain and cross-domain data demonstrates the superior mapping accuracy and generalizability of SGMap-Net over state-of-the-art geologic multimodal fusion models.
>
> # "2. Incomplete Baseline Comparison"
>
> In addition to the experiments presented in the paper, we conducted exploratory evaluations of several SOTA multimodal foundation models, including SatMAE and SatMAE++, as well as DOFA, and Panoptic-FM (see our response to Reviewer Fx2N). The performance of SatMAE and SatMAE++ is included in Sec. 3.4 and Table 15 (Supp.). We have just recently evaluated DOFA and Panoptic-FM models, per the suggestion of ReviewerFx2N, however, these models were originally designed for spectral (I.e., RGB+NIR) remote sensing tasks and did not perform competitively in our setting. In contrast, ES emphasizes non-spectral geospatial modalities such as slope, elevation percentile, and other terrain-derived features that capture shape-based information critical for surficial geologic mapping. SGMap-Net was explicitly designed to integrate these modalities through a fusion architecture tailored for geologic context, and it consistently outperformed the general-purpose models in both in-domain and cross-domain settings. As we are currently moving into segmentation experiments, we will certainly be using additional baseline comparisons relevant to that task, including the ones you mentioned.
>
> # "3. Dataset Contribution is Overstated for the ML Community"
>
> While we understand that ES may initially appear modest in terms of class count and sample size relative to large-scale vision datasets, we respectfully suggest that this framing overlooks several key contributions of the dataset that are directly relevant to the ML community. ES is not simply a patch-level classification dataset, but instead presents a complex, high-dimensional, and multimodal learning challenge with strong grounding in spatial reasoning, fusion architecture design, and real-world generalization.
>
> Although ES v1.0.1 contains 31,066 patches, it is a modular and extensible benchmark by design. Fifteen additional areas are already queued for release, which will add approximately 62,000 new patches and nearly triple the dataset patch count. These additions will introduce greater geologic, anthropogenic, and local environmental diversity, expanding the range of modeling tasks and enabling broader generalization research across geologic settings.
>
> Each ES patch includes 38 fully co-registered input channels, including high-resolution 4-band RGB+NIR imagery, a DEM, five terrain derivatives each calculated at six different spatial resolutions to capture spatial hierarchies, rasterized vector data for infrastructure and hydrology, and expert-labeled segmentation masks, labels, and class areas. Unlike typical remote sensing datasets that rely on overhead imagery, ES presents a fusion task involving diverse modalities that reflect fundamentally different physical processes. This configuration enables ML research on cross-attention mechanisms, modality dropout, multi-scale representation learning, and efficient transformer architectures to process the dataset.
>
> ES supports multiple task formulations beyond classification. The dataset includes per-patch class proportions for multi-output regression and high-resolution segmentation masks for pixel-wise modeling. These features allow researchers to explore semantic segmentation, boundary detection, domain adaptation, and spatial generalization. The current best-performing model, which fuses elevation percentile, slope, and roughness using early channel stacking, achieves a macro-F1 of 0.657 in-domain and 0.598 on the held-out county. These results are far from saturated and suggest considerable room for improvement in both architectural design and training methodology.
>
> ES also presents a class distribution that is both long-tailed and spatially entangled. Units such as artificial fill (af1) and alluvial fans (Qaf) are rare, but are crucial for hazard and infrastructure applications. Classes frequently co-occur within individual patches, creating a highly imbalanced multilabel problem where class proportions can vary from near-zero to complete coverage. This poses a demanding test case for rare-class detection, long-tail learning, and loss function tuning, none of which are trivial in geospatial settings. ES also provides a dataset for the ML community to explore domains that exhibit spatial autocorrelation, hierarchical structure, semantic ambiguity, and incomplete boundaries, all of which are central challenges in real-world scientific mapping tasks.
>
> While ES is not large in raw sample count compared to many vision datasets, its design is grounded in real-world data complexity, physical relevance, and computational challenge. It provides a platform that bridges the gap between ML methodology and practical application. We will work to make these points more explicit in the camera-ready version.
>
> # "4. Limited Geographic Scope and Generalizability"
>
> We agree that geographic diversity is essential for evaluating model generalization, and we refer you to our response to Reviewer PsPk as well as the summary provided to Reviewer Fx2N. While ES v1.0.1 is currently limited to two counties in central Kentucky, these regions exhibit meaningful geomorphic contrast, a measurable domain shift, and robustness to temporal discrepancies across modalities. The geologic processes represented by the mapped target classes are driven by universal surface-forming processes that are not tied to local geography. Our experiments demonstrate that the best-performing multimodal model yields a cross-domain macro-F1 drop of just 0.059, compared to a drop of 0.271 for the best-performing single-modality baseline (Table 1). This indicates that domain shift can be substantially mitigated through the use of multimodal inputs. We also show that elevation-derived terrain features improve both in-domain and cross-domain performance, supporting the idea that shape-based features are physically grounded, region-invariant, and more robust to unseen settings. Model performance also remains stable across regions despite potential differences in data acquisition periods and the possibility of localized geologic change, though we don’t expect these factors to be significant. Finally, our GitHub is well-documented and the dataset preprocessing pipeline is straightforward for users to expand ES to include additional regions. ES is currently expanding to include approximately 62,000 new patches, each containing 38 input channels.

---

> > ### Comment · Reviewer_3sWo · 2025-08-04
> >
> > Thank you for the detailed and thoughtful response.
> >
> > I acknowledge the authors’ effort in constructing the ES dataset and designing SGMap-Net to suit the specific challenges of multimodal surficial geologic mapping. The inclusion of terrain-derived features, multimodal fusion, and cross-domain evaluation does reflect careful consideration and technical depth.
> >
> > That said, I remain unconvinced that the current version of the dataset offers a sufficiently high-impact contribution for the broader ML community, especially at the scale expected for a NeurIPS submission. While the authors emphasize the complexity and extensibility of ES, it is worth noting that many established datasets in computer vision and even in medical imaging—e.g., those used for segmentation, multi-modal fusion, or rare-class detection—are larger in scale and support a broader array of tasks, often with clearer benchmarking protocols and wider adoption potential.
> >
> > Indeed, almost any sufficiently rich dataset can be extended to support multiple tasks, and the fact that ES allows for classification, regression, and segmentation is not in itself exceptional. The authors’ argument that ES is designed for spatial reasoning and physically grounded learning is appreciated, but such properties are not uncommon in existing remote sensing datasets or even in urban scene understanding tasks with LiDAR, DEM, and vector data.
> >
> > In short, while I recognize that the dataset and model are thoughtfully constructed and may be valuable within their specific domain, I believe the dataset’s scale, maturity, and broader utility fall somewhat short of the bar typically expected for a dataset-centric contribution at NeurIPS. My overall assessment remains unchanged.

---

> > > ### Author Response · Authors · 2025-08-06
> > >
> > > Thank you for your thoughtful follow-up. We appreciate your acknowledgement of the care taken in constructing both the EarthScape dataset and the SGMap-Net architecture. Below, we offer a few final points to reinforce the broader relevance and design intentions of this work.
> > >
> > > ***1. Scale & maturity.*** We agree that ES v1.0.1 is smaller in geographic coverage than some large-scale CV datasets. However, it provides 1230 <sup>2</sup> of labeled data (e.g., 5x the labeled area of SpaceNet-6) with 38 co-registered channels per patch and dense segmentation masks. *What sets ES apart is not the sample count, but its modality complexity, physical grounding, and real-world difficulty*. We show that even strong foundation models using typical RS modalities underperform when tested cross-domain, whereas SGMap-Net narrows this gap considerably. This is neither trivial nor saturated. Growth is active, with v1.0.2 (Q4 2025) tripling the sample count and v1.0.3 expanding out-of-region. ES is new, novel, and growing.
> > >
> > > ***2. Controlled Multi-Task Benchmarking.*** Unlike existing RS datasets that allow multiple tasks, ES supports unified benchmarking across classification, segmentation, and regression, all grounded in the same high-resolution, physically interpretable inputs.
> > >
> > > ***3. Broad ML Utility.*** ES directly supports general-purpose ML challenges including multimodal fusion, rare-class detection, long-tailed multilabel learning, modality dropout, and domain generalization, challenges that extend well beyond geology. Its shape-centric design enables testing of physically grounded representation learning, which is increasingly relevant to scientific ML, environmental modeling, and disaster forecasting. *While ES may not resemble the typical large-scale image datasets, it introduces a unique spatial, physical, and multimodal challenges that are underrepresented in current benchmarks and valuable to the broader community.*
> > >
> > > ***4. Alignment with NeurIPS D&B Scope.*** ES is a carefully constructed, open-source AI-ready dataset with reproducible benchmarks and a modular pipeline for community expansion. It aligns with the D&B goals of:
> > > * “New datasets…carefully and thoughtfully designed”
> > > * “Data-centric AI methods and tools…that bring important new insight”
> > > * “Benchmarks on new or existing datasets…”
> > >
> > > We hope this clarifies ES’s broader relevance. We sincerely appreciate your engagement and the improvements your comments have helped us make.

---

### Official Review · Reviewer_99ix · 2025-06-30

**Rating:** 4
**Confidence:** 4

**Summary:**

Surficial geologic maps are essential for understanding Earth surface processes, but large-scale mapping remains labor-intensive and costly. To address this, the authors introduce *EarthScape*, a multimodal geospatial dataset that includes DEMs, remote sensing imagery, and GIS vector data. They formulate surficial geologic mapping as a patch-level multi-label classification task and propose a baseline model, SGMap-Net.

**Additional Feedback:**

- The authors are encouraged to specify the most suitable configuration of SGMap-Net, such as the preferred strategies for multi-scale and multi-modal fusion.
- The evaluation includes both CNN- and Transformer-based architectures. It is suggested to summarize their respective strengths and applicability to SG mapping tasks.
- It would be helpful to provide a visual comparison between the predicted results and the ground truth in the form of maps, to better illustrate the gap between current AI methods and human expert interpretations.

**Dataset Code Accessibility:**

Yes

**Ethical Considerations:**

No, there are no or only very minor ethics concerns

**Final Justification:**

The authors have effectively addressed most of my concerns. I also believe that the dataset can make a valuable contribution to advancing geoscience with AI. As such, I am happy to raise my score.

**Limitations Weaknesses:**

- From Supplementary Table 10, the proposed attention-based cross-modal fusion mechanism appears ineffective, as it even leads to performance degradation. The best-performing configuration (EP+S+SDS) instead relies on early-stage channel stacking for fusion.
- While GIS vector data are introduced in the *EarthScape* dataset and are claimed to benefit SG mapping, their contribution is not explicitly evaluated in the SGMap-Net experiments.
- It is recommended that the authors further clarify the fusion strategies used for multi-scale fusion, particularly the attention-based approach. It would be helpful to specify whether this differs from the strategies used for multimodal fusion.
- The choice of kernel sizes (5×5, 11×11, 21×21, 51×51, 101×101, 185×185, and 201×201 pixels) for computing terrain derivatives should be justified.
- The authors are also encouraged to explain why some ViT results are missing in Supplementary Tables 10, 11, and 12.
- The paper should clarify how multi-labels are assigned to each patch—for example, the minimum area proportion required for a class to be considered a label—and justify the rationale behind this choice.

**Strengths Contributions:**

- The *EarthScape* dataset integrates DEMs, multispectral remote sensing imagery, and GIS vector data. Domain knowledge is leveraged to handcraft DEM-based features, resulting in 38 geologically meaningful channels per patch to facilitate surface representation learning.
- A baseline model, SGMap-Net, is introduced to formulate SG mapping as a patch-level multi-label classification task, enabling the prediction of seven geologic classes.
- Comprehensive experiments with representative CNN and Transformer backbones (ResNeXt and ViT) are conducted to evaluate the contributions of different geospatial modalities, as well as their combinations and fusion strategies. The study further investigates the impact of multi-scale terrain derivatives and introduces a cross-region evaluation protocol to account for geographic heterogeneity.

---

> ### Author Rebuttal · Authors · 2025-07-31
>
> We appreciate the critique and curiosity from Reviewer 99ix, and for recognizing the richness of EarthScape (ES), and the depth of our baseline experiments and evaluation protocol.
>
> # "Supplementary Table 10"
>
> Our primary objective in this study was to evaluate various fusion strategies for effectively integrating multiple modalities within the proposed ES dataset to classify geographic categories. To this end, we conducted a comprehensive set of experiments exploring different fusion configurations, including early-stage stacking, concatenation, and mid-level attention-based mechanisms such as cross-modal attention. Our empirical results indicate that early-stage stacking consistently yields superior performance in this setting. For example, using the ResNeXt-50 backbone, the configuration with EP+S+SDS (early stacking) achieved the highest in-domain F1 score of 0.657, while the attention-based counterpart (shared encoder) reached only 0.561. We believe this is due to the strong spatial and semantic alignment between modalities in ES, which makes early fusion more effective at capturing complementary information. Although attention-based fusion did not outperform early stacking in our current setup, we plan to explore more fusion strategies in future work.
>
> # "Contribution of GIS vector data"
>
> We appreciate the reviewer highlighting the omission of any experiments with the GIS vector layers, OpenStreetMap (OSM) infrastructure and U.S. Geological Survey National Hydrography Dataset (NHD) hydrography. These layers were included in ES because they are frequently used by experts to aid interpretation of artificial fill (af1), which often coincides with major road and rail infrastructure, and alluvial deposits (Qal, Qaf, Qat), which are typically found near stream centerlines. Our original intention was for these layers to complement terrain and imagery inputs in identifying these specific units, rather than as single modality inputs. However, we have processed results for single modality tests to use as baselines and include selected results here (global metrics with the ResNeXt backbone):
>
> | Modality | Accuracy | Macro Precision | Macro Recall | Macro F1 | Macro mAP |
> |-----------|------------|--------------------|-----------------|------------|----------------|
> |NHD | 0.682 | 0.419 | 0.725 | 0.515 | 0.403 |
> OSM | 0.647 | 0.442 | 0.846 | 0.530 | 0.435 |
>
> Comparison with Table 4 (Supp.) shows that these features underperform compared to many features (maximum macro-F1 of 0.651), but still provide slightly more discrimination compared to planform and profile curvatures (minimum macro-F1 of 0.488). Our ongoing effort tests their contributions to multimodal fusion models. We will report these results in the camera-ready version to provide a more complete picture of the dataset and feature contributions.
>
> # "multi-scale fusion"
>
> The attention-based approach used for multi-scale fusion is architecturally identical to the one used for multimodal fusion. In both cases, the architecture follows the design outlined in Sec. 4.1. The only difference is the input domain: for multimodal fusion, attention is applied across different sensor modalities (e.g., RGB, DEM, terrain features), while for multi-scale fusion, the same mechanism is applied across different spatial resolutions of a single modality (e.g., S computed at resolutions of 5, 10, 20, etc. ft/pixel). We tested whether attention-based fusion of multiple spatial resolutions within a modality could improve classification by capturing features associated with geologic processes expressed at different spatial scales. This design draws on established geological understanding that surface processes operate across a range of scales, and the core objective here was to assess whether fusion of scale-specific signals improves model performance. In-domain performance gains were modest, but models trained with multi-scale fusion showed improved generalization under domain shift, suggesting that multi-scale context increases robustness. This is discussed in Sec. 4.2 and detailed in Tables 7–9 (Supp.).
>
> # "choice of kernel sizes"
>
> The specific sizes were chosen to provide a consistent and interpretable representation across a range of spatial resolutions, where each scale is expected to capture different land-surface processes or geologic features. For S, profile curvature (PrC), and planform curvature (PlC), we first downsampled the native 5 ft/pixel DEM to coarser resolutions (10 ft, 20 ft, 50 ft, 100 ft, and 200 ft/pixel), then computed each derivative using a fixed 5×5 kernel at each resolution. This downsampling strategy provides a progression of spatial scales, allowing models to learn geomorphic patterns ranging from local landform texture to broader topographic structure. The selected resolutions follow a roughly logarithmic progression, which is common in geomorphometric analysis when no "correct" scale exists.
>
> EP and SDS are inherently neighborhood-based statistical features rather than derivatives of elevation. These were calculated using neighborhood-based statistics, where we retained the original 5 ft/pixel resolution, but varied the kernel size directly. The selected kernel sizes, 5×5, 11×11, 21×21, 51×51, 101×101, and 201×201 pixels, were chosen to mirror the physical footprint of the downsampled DEMs used for S, PlC, and PrC. This alignment ensures that each scale of EP or SDS reflects a neighborhood size comparable to one of the spatial resolutions used for S, PrC, or PlC, allowing direct comparison across modalities. For example, EP computed with a 101×101 kernel covers approximately the same area as slope derived from a 100 ft/pixel DEM.
>
> While there is no standardized set of kernel sizes in geologic mapping, these choices were designed to span a representative range of spatial scales relevant to surficial processes without introducing unnecessary redundancy. Our experimental results indicate that no single scale performs best across all classes, and that combining multiple scales improves cross-domain robustness. We will clarify this rationale in the camera-ready version.
>
> # "missing ViT results"
>
> Thank you for bringing this to our attention. The ViT results were unintentionally omitted from Tables 10-12 (Supp.) in the original submission. We have now included the missing ViT results.
>
> # "assigning multi-labels to each patch"
> We agree that label assignment is a critical design decision, especially in multilabel patch-based benchmarks. In ES, labels were assigned via spatial overlay between patch polygons and vector-format surficial geologic units. If a geologic unit intersects a patch, even marginally, its class is included as a label, with no minimum area threshold. This policy was adopted to ensure complete label coverage, however, our dataset includes class area proportions per patch allowing users to easily customize their labeling strategies using any size threshold. We will include a clearer explanation of the labeling procedure and its implications in the camera-ready version for evaluation to ensure transparency.
>
> # "most suitable configuration of SGMap-Net"
>
> For multi-modal fusion, the best-performing configuration used early channel stacking of EP, S, and SDS, each at a single spatial scale, with a ResNeXt-50 backbone. This configuration achieved the highest in-domain F1 score (0.657) and also generalized well to the held-out region, with a cross-domain F1 of 0.598 (see Table 1 and Supp. Tables 10-12). For multi-scale fusion, early fusion (via channel stacking) consistently outperformed attention-based fusion. We observed that combining multiple spatial scales of a single terrain feature (e.g., slope computed at 5, 10, 50, 100, 200 ft/pixel) improved generalization, though it did not always improve in-domain accuracy (see Table 1 and Supp. Tables 7-9). Across all fusion experiments, attention-based fusion generally underperformed relative to early channel stacking and mid-level concatenation strategies.
>
> # "strengths and applicability of CNN- and Transformer-based architectures"
>
> In our experiments, CNN-based models (e.g., ResNeXt-50) consistently achieved higher in-domain performance, particularly when using shape-derived terrain features such as S, EP, and SDS. This suggests that CNNs are well-suited to capturing localized spatial patterns and fine-grained geomorphologic cues that are critical in surficial geologic map interpretation. Transformer-based models (e.g., ViT-B/16), on the other hand, demonstrated slightly better robustness in cross-domain generalization, especially in scenarios involving less structured inputs or weaker visual signals.
>
>
> # "visual comparison between the predicted results and the ground truth"
> This is an excellent suggestion! In the camera-ready version, we will include a map-based visualization that shows how patch-level classifications aggregate across space.

---

> > ### Comment · Reviewer_99ix · 2025-08-07
> >
> > The authors have effectively addressed most of my concerns. I also believe that the dataset can make a valuable contribution to advancing geoscience with AI. As such, I am happy to raise my score.

---

> > > ### Author Response · Authors · 2025-08-08
> > >
> > > Thank you for your feedback and for recognizing the contribution of our dataset! We appreciate all your thoughtful suggestions and questions, which have helped us improve the quality and clarity of our paper.

---

### Official Review · Reviewer_Fx2N · 2025-06-30

**Rating:** 5
**Confidence:** 5

**Summary:**

This paper introduces a new multi-modal, multi-scale ML benchmark dataset for surficial geologic mapping and Earth surface analysis. It also introduces the first set of deep learning baselines on this benchmark, opening the opportunity for future competitions. Although the dataset is derived solely from data sources in Kentucky, it targets a relatively underexplored application of remote sensing and machine learning that desperately needs more ML benchmark datasets. My current recommendation is "borderline accept", but there are a few low-hanging fruit that can be addressed during the rebuttal phase that would increase my score to "accept".

**Additional Feedback:**

Scales (e.g., 1:24,000) are commonly used by geologists, but ML audiences may be unfamiliar with this terminology. Also, scales don't necessarily make sense in the age of digital maps where anyone can zoom in or out. It would improve accessibility if you provide some kind of explanation of scales, or even switch to ground sample distance (GSD) in meters, as this is more commonly used in the remote sensing community.

For international publication, supplementing or replacing imperial units (in/ft) with metric units (m/cm) would be very appreciated.

The dataset contains 38 channels, but these channels aren't well explained. I'm guessing there are 4 channels of RGB-NIR imagery and 1 channel of DEM, but I don't know the rest.

Float32 may not be necessary, uint8 is commonly used to save space and may be sufficient for this task, especially for the RGB-NIR imagery and class labels/masks.

GeoJSON files may be difficult for ML folks to process, it may be worth including a rasterized version for easier access.

In mathematical equations, words like "encoder" or "softmax" should not be italicized.

* Line 274: the degree symbol is not correct
* Line 293: missing period

**Dataset Code Accessibility:**

Yes

**Dataset Code Comments:**

In order to actually use your dataset or code, it must be distributed under a license that permits users to use it. The GitHub repository currently has no license, and the paper mentions no license either. The University of Kentucky website mentions CC-BY-4.0, which is a good license for the data, but doesn't really make sense for the code. Typically, GitHub is only used to store code, and sites like Hugging Face or Zenodo are used to store the actual data. Also, you can safely remove files like .DS_Store or __pycache__ directories from GitHub, as they are not needed for reproducibility and can actually cause issues when .pyc files are transferred between Linux and macOS.

It may be worth adding your data loaders to a library like TorchGeo (https://github.com/microsoft/torchgeo) to make it easier for ML researchers to experiment on your dataset in a single common workflow. TorchGeo already provides a variety of builtin datasets (https://torchgeo.readthedocs.io/en/stable/api/datasets.html), including many for earthquake detection and flood mapping. However, this reviewer may or may not be affiliated with TorchGeo, and so my review score is independent of this decision.

**Ethical Comments:**

Only minor ethical concern is that the data is only representative of a small region in Kentucky, but this is somewhat mitigated by inclusion of a geographic train-val-test split for out-of-domain generalization testing. It may be worth adding a sentence clarifying that users of this dataset or model should not generalize to completely new regions without careful validation.

**Ethical Considerations:**

No, there are no or only very minor ethics concerns

**Final Justification:**

This paper introduces one of the first ML-ready benchmark datasets for long-term surficial geology. This is an incredibly valuable benchmark for both the surficial geology and remote sensing communities, and was created with very careful attention to detail and interdisciplinary contributions.

My comment on the lack of semantic segmentation masks turned out to be a misunderstanding. The authors have promised to update the Related Work section to better reflect more recent geologic datasets and clarify the differences between ES and these more short-term datasets. I would still argue that SeafloorAI is the first long-term surficial geology dataset (albeit limited to the seafloor), but I agree that the other datasets I mentioned are for a completely different subdomain within geology. Similarly, while I still find SGMap-Net to be a very primitive multi-modal architecture, I appreciate the authors' experimentation with more modern multi-modal foundation models. All comments with regards to dataset and code availability have been addressed.

While I agree with other reviewers that this dataset is likely not very useful to the broader ML community, I stand by my point that this dataset is incredibly valuable to my own research community in geology and remote sensing. As such, I have increased my rating to "Accept".

Even if this paper is not accepted at NeurIPS, I hope to see it published elsewhere, as it is a fine work.

**Limitations Weaknesses:**

There are several easy-to-resolve weaknesses that I think the authors should focus on during the rebuttal phase:

**Related Work**: While the Introduction is quite good, the Related Work section could use some work. In particular, the section on Remote Sensing Datasets lacks any datasets from the last 6 years! While I agree with the authors that there are very few surface geology datasets, I don't believe this is the only one. For example, https://github.com/RichardScottOZ/mineral-exploration-machine-learning lists many potential data sources (some ML-ready, some not), and https://arxiv.org/abs/2411.00172 offers many of the same features but for seafloor geology. There are several other geologic datasets out there (https://arxiv.org/abs/2403.18116) but these tend to be limited to detection of earthquakes, landslides, and other natural hazards. Similarly, there are many other hydrologic datasets out there (https://doi.org/10.1109/ACCESS.2022.3205419, https://nasa-impact.github.io/etci2021/) but these tend to be limited to flood detection. Instead of the general purpose land surface mapping datasets currently cited, I would suggest citing some of these more closely related works.

**Semantic Segmentation Masks**: It seems like the dataset is currently limited to multilabel classification. This is very unfortunate, as it seems like semantic segmentation masks are readily available. I would encourage the authors to release these masks directly as a semantic segmentation challenge. Semantic segmentation is significantly more useful to both remote sensing and geologic mapping, and classification tends to be a bit too easy of a challenge.

**Multi-Modal Models**: The SGMap-Net model introduced in this work is quite primitive, especially the use of a 1x1 convolution to standardize input to 3 channels. In the last couple of years, there has been an explosion of multi-modal foundation models designed to incorporate inputs from any sensor, with any spatial resolution, and for any number of input channels. Some models like DOFA (https://arxiv.org/abs/2403.15356) and Panopticon (https://arxiv.org/abs/2503.10845) are limited to spectral images (SAR, RGB, MSI, HSI), but some newer models like Copernicus-FM (https://arxiv.org/abs/2503.11849) or TerraMind (https://arxiv.org/abs/2504.11171) can directly incorporate additional metadata like DEMs. Instead of introducing a new model, it may be better to benchmark existing multi-modal models.

The license should also be clarified, see "Dataset Code Comments" below.

There are also some more challenging limitations that I don't expect the authors to be able to address during rebuttal.

**Geographic Scope**: As mentioned in the Limitations section, the dataset is currently limited to a small region in Kentucky due to data availability. While this isn't ideal, the inclusion of a geographic split helps to evaluate out-of-domain performance.

**Temporal**: While the authors mention temporal inconsistency between layers, I'm actually more concerned about the lack of temporal imagery. Although many things like geologic and soil properties are mostly static, some things like hydrologic properties are very dynamic and will change throughout different seasons. In future versions of this work, inclusion of time series imagery from Sentinel-2 or even hyperspectral data may help improve model performance.

**Strengths Contributions:**

The greatest strength of this paper is the interdisciplinary team that created this benchmark dataset. Without experts from both geology and computer science, this level of quality would not be possible. The care with which the dataset is documented, including data provenance and preprocessing, out-of-distribution (domain adaptation) train-val-test splits, and benchmark experiments, set a standard for other geospatial ML datasets.

The second greatest strength of this work is the relative scarcity of benchmark datasets for this task. As the authors point out, existing benchmarks like ImageNet and COCO have led to great progress in computer vision research. Without more benchmarks for surficial geology and hydrology, this kind of progress is not possible.

The authors use common easy-to-use file formats with geospatial metadata that can easily be read by GDAL/rasterio/fiona/geopandas. This geospatial metadata can also be directly incorporated into models, which may reduce transferability to new geographic regions, but can also take advantage of known geographic differences between soil and rock distributions.

---

> ### Author Rebuttal · Authors · 2025-07-30
>
> We thank Reviewer Fx2N for the detailed and encouraging feedback, recognizing the interdisciplinary foundation of the work, the care taken in in our documentation, and the importance of EarthScape (ES) as a much-needed benchmark for geospatial ML.
>
> # "Related Work"
>
> These are great resources and we appreciate your suggestions. In the camera-ready version, we will revise Sec. 2 to include and contextualize the datasets mentioned. We would also like to expand on your point about how ES differs from these efforts. Many geologic datasets, including those cited, focus on short-lived transient events (flooding, earthquakes, or slope failures), rather than stable Earth materials. In contrast, ES targets the underlying unconsolidated geologic units that govern surface processes and geotechnical behavior controlling these events. We see ES not as an alternative to these datasets, but as a complement that offers additional geological context. Our dataset can be used on its own or to enrich related studies like flood modeling, landslide mapping, and mineral prospectivity workflows by anchoring predictions in the physical materials that shape the landscape.
>
> # "Semantic Segmentation Masks"
>
> We wholeheartedly agree about the importance of semantic segmentation for both remote sensing and geologic mapping, however, segmentation masks are already included in ES. These masks are shown in Fig. 1, discussed in Sec. 3.2, and displayed in Figs. 3 & 4 (Supp.); the dataset README and DataDictionary also contain information on available channels, including the masks. We also include class labels and per-class patch proportions, but the full pixel-wise segmentation masks are also provided directly as GeoTIFFs for each patch. In our paper, we focused on multilabel classification as a baseline task due to its simplicity, tractability, and relevance for coarse-grained geologic model evaluation. However, this was not due to any limitation in the dataset itself. ES was designed from the outset to support semantic segmentation. We are currently building on our classification results and transitioning to benchmarking the segmentation task.
>
> # "Multi-Modal Models"
>
> To clarify, SGMap-Net was introduce as a lightweight, transparent architecture to establish baseline performance across input combinations. Its simplicity allows users to isolate the contributions of modality, scale, and fusion strategy without architectural confounds. We did test SatMAE and SatMAE++ (Supp. Sec. 3.4, Fig. 15), which both underperformed compared to SGMap-Net.
>
> That said, we fully agree that benchmarking against more advanced multimodal models is important. Per your suggestions, we have already evaluated DOFA and Panopticon using RGB+NIR inputs. These models also underperformed relative to SGMap-Net variants that incorporated terrain-based modalities, which we expected, as overhead imagery does not capture surficial material properties in the same way that terrain derivatives do. We are also currently exploring Copernicus-FM. All new baseline experiments will be included in the camera-ready version.
>
> # "Geographic Scope"
>
> As noted in our detailed response to Reviewer PsPK, we agree that broader geographic coverage is essential for evaluating model generalization. While the initial release of ES includes only two counties in central Kentucky, these regions enable meaningful cross-domain evaluation. First, there is a measurable domain shift between them, and our experiments show that multimodal inputs improve model robustness under this shift. Second, the geologic units included (Qal, Qaf, Qat, Qc, Qca, Qr) reflect universal surface processes (i.e., fluvial deposition, hillslope transport, in-situ weathering) and are not geographically unique to the study area. Third, our benchmarks show that models trained on terrain-derived surface features consistently outperform those using raw elevation or imagery, underscoring their value for generalizable geologic representation. Fourth, the two regions were mapped in different time periods (Hardin County: 2018–2022; Warren County: 2022–2024), yet our models generalize well across this temporal offset, suggesting resilience to variation in acquisition date, urban expansion, or infrastructure growth. Finally, ES is designed for modular expansion, and we have already queued 15 additional quadrangles for release, which will nearly triple the number of patches and introduce greater lithologic, geomorphic, and climatic diversity. We appreciate your and Reviewers PsPK and 3sWo’s recognition of the geographic split’s importance, and will emphasize these points more clearly in the camera-ready version.
>
> # "Temporal"
>
> ES is designed to map surficial geologic materials that are largely stable over geologic timescales (10³–10⁵ years), and their delineation is primarily governed by terrain shape, relative elevation, and geomorphic context. Accordingly, ES v1.0.1 emphasizes static, physically interpretable modalities such as DEM-derived terrain features, aerial imagery, and vector GIS data. That said, we agree that temporal or multi-temporal imagery may provide valuable contextual signals, especially for distinguishing materials based on hydrologic, mineralogical, or vegetation-related properties. While time series inputs are not typically used by geologists in surficial mapping workflows, we recognize their potential utility, particularly in future extensions that target more dynamic features such as landslides, sinkholes, or seasonal fluvial changes. Our existing patch structure and data pipeline are compatible with the integration of time series or hyperspectral inputs. While such data is not included in the current version, we are actively exploring its incorporation as part of future development. We also refer you to our response concerning a similar matter from Reviewer PsPK.
>
> # "minor ethical concern"
>
> We agree that models trained on ES should not be assumed to generalize to new regions without careful validation and expert interpretation. Surficial geologic units are inherently tied to lithology, topography, and climate, and their expression may differ significantly across physiographic provinces. However, one of ES’s goals is to support research into geographic generalization. Our baseline experiments show that models using terrain shape (e.g., slope, curvature, elevation percentile) exhibit greater robustness under domain shift than models trained on overhead imagery or raw elevation alone. This is encouraging, and we are actively investigating how these features transfer across regions outside of the current ES geographic scope. We will add a clarifying statement in the camera-ready version (Sec. 5).
>
> # "Dataset Code Comments"
>
> We have added an MIT License to the GitHub. We have also removed .DS_Store and __pycache__ files. We will also explore contributing our dataset loader to TorchGeo.
>
> # "Scales (e.g., 1:24,000)"
>
> We agree this clarification would improve accessibility and we recognize that this terminology may be unfamiliar to many in the ML and RS communities. In the camera-ready version, we will report GSD (meters) whenever referring to raster resolutions. When referring to the original geologic maps, we will retain the 1:24,000 scale notation (as it defines the intended cartographic resolution), but will add an explanation of this concept to help bridge disciplinary terminology.
>
> # "replacing imperial units (in/ft) with metric units (m/cm)"
>
> We will revise the camera-ready version to use metric units as the primary system.
>
> # "channels aren't well explained"
> The dataset channels are detailed in Sec. 3, Sec. 2.4 and Figs. 3 & 4 (Supp.), and the README and DataDictionary files downloaded with the dataset. That said, we will revise the camera-ready version for clarification. We have also added a list of available ES images on our GitHub.
>
> # "uint8 to save space"
>
> We agree that float32 is not necessary for all modalities (RGB+NIR, hydrography and infrastructure layers, and segmentation masks). These were originally kept in float32 for consistency across preprocessing, but we will convert appropriate layers to uint8 in the next dataset release (ES v1.0.2).
>
> # "GeoJSON files"
>
> The GeoJSON files are primarily used for dataset curation and pre-processing. ES does include GeoJSON files for geolocation of individual patches, but our GitHub already includes documented code to handle all patch selection, label assignment, and data extraction based on the GeoJSON definitions. Also, since patches are partially overlapping by design, they are not well-suited for rasterization into a single, non-overlapping grid.
>
> # “Grammatical”
>
> We have fixed the typos and words in equations, thanks for pointing to those.

---

> > ### Comment · Reviewer_Fx2N · 2025-08-05
> >
> > I would like to thank the authors for taking the time to write a detailed response to all of my comments. All of my concerns and limitations are either being addressed or were due to my own misunderstanding about the dataset. While I still believe that SGMap-Net is a bit primitive in model architecture, I understand that existing multi-modal foundation models either don't support non-image modalities or are challenging to fine-tune on such a different dataset. I look forward to seeing the new analysis provided by the authors.
> >
> > While I agree with other reviewers that this dataset is likely not very useful to the broader ML community, I stand by my point that this dataset is incredibly valuable to the surface geology and remote sensing communities. Even if this paper is not accepted at NeurIPS, I hope to see it published elsewhere, as it is a fine work. I have updated my review score to "Accept".

---

> > > ### Author Response · Authors · 2025-08-06
> > >
> > > Thank you for revisiting the paper, the updated score, and your encouragement! We agree the dataset is mid‑scale, yet its rich set of features (raster + vector channels) and demonstrably hard cross-generalization split offer a unique benchmark for multimodal fusion.

---

### Official Review · Reviewer_PsPK · 2025-07-01

**Rating:** 4
**Confidence:** 3

**Summary:**

This submission presents Earthscape, a dataset for geologic mapping and Earth surface analysis. The dataset provides several modalities that can be divided into four groups: the surficial geologic maps (multilabel targets and segmentation masks, and vector polygons), remote sensing data ( aerial imagery and Digital Elevation Models (DEM)), DEM-derived features for the geomorphometric analysis of the terrain, and hydrographic/infrastructure features. The dataset is articulated around two regions in Kentucky, U.S. with a data acquisition campaign between 2019 and 2024 (for images, DEM and vector maps). Besides the dataset, this submission also proposes benchmarks for evaluating models in different setups on a multilabel classification task (with 7 geological classes).

The main contribution of this work probably lies in the richness of this dataset that mixes geologic and remote sensing data. Notably, this work also proposes a set of features derived from the raw data, as well as a fusion-based model and benchmarks that analyse the impact of this dataset on classification models with a good number of nteresting experiments.

**Dataset Code Accessibility:**

Yes

**Dataset Code Comments:**

The dataset code seems well accessible and consistent with the content of the paper. The readme file is concise, but gives the necessary info for using and exploiting the dataset.

**Ethical Comments:**

i did not identify potential ethical concerns.

**Ethical Considerations:**

No, there are no or only very minor ethics concerns

**Final Justification:**

The rebuttal and the clarifications from the authors were mostly convincing. They gave clear answers to my concerns on the stability in time and on the topic importance/application fields. The restricted coverage of the dataset is probably, to me, the weak point of the dataset. That said, I beleive it remains valuable in its current state. I encourage the authors to extend the dataset with new geographical zones in the future.

**Limitations Weaknesses:**

- one weaknessis probably the restricted  coverage of the dataset. It only consists in two aeras, originating from the same region in the world (the state of Kentucky in U.S.. This lack of geographic diversity might not be impactfull for measuring the importance of data modalities and features on the geological classification tasks. However, it is quite penalizing in the perspective of designing generic classification algorithms in the field. At least the generalization aspects are not discussed enough. Calling the dataset EarthScape is also probably not very appropriate as it gives the feeling of global coverage at the scale of the Earth.

- The submission does not clearly explain the importance of the addressed topic for research or industry. At least, it is not clearly shown the dataset will benefit to a potentially large research community working in the field, or will open the door to the design of new methods and the realization of new studies. That said, i am not a geologist and cannot evaluate this point properly. I hope authors (in the rebuttal) and other reviewers will remove my doubts on that.

- The fact the dataset is composed of modalities acquired during a five year period might be problematic. It would have been interesting that the authors dicuss the stability in time of the two zones, e.g. do natural phenomena of potential urban extensions could have modified the terrain during the five years ?

**Strengths Contributions:**

- Richness of the dataset in terms of modalities/features. The dataset goes beyond the existing ones by mixing traditional geologic maps with remote sensing data and specific features. The dataset includes pixel-based information, but not only. Vector-based representations of some objects and classes are also given.

- The paper is well-written and nicely organized. The reading is easy and fluid. The design choices are also globally well motivated.

- The technical content seems to be coherent. In particular , I did not notice any particular problem in the experimental validation. The provided materials (code and dataset) seem to be consistent.

- The experimental validation presents a good number of interesting ablations for analyzing the modality/feature importance.

---

> ### Author Rebuttal · Authors · 2025-07-30
>
> We thank Reviewer PsPk for the thoughtful and well-reasoned feedback, and for recognizing the richness and diversity of the dataset and the strengths of our experiments.
>
> # "restricted coverage of the dataset"
>
> We agree that geographic diversity is essential for evaluating generalization in ML, and that EarthScape (ES) is currently limited to two counties. However, we respectfully submit that the dataset, even in its current form, offers scientifically meaningful challenges for ML.
>
> **Geographic Domain Shift:** Both Counties in ES represent contrasting topographic settings. Although class balance is similar in both, the dominant geologic processes are different. Warren County is characterized by low-relief karst plains and dominated by in-situ weathering, whereas Hardin County comprises a dissected landscape with predominantly erosional processes. In practice, these contrasts are sufficient to induce a measurable domain shift: our best single-modality model exhibits a 0.271 macro-F1 drop under cross-domain evaluation, while our best multimodal model shows a significantly smaller decline of 0.059. This gap illustrates both the value of multimodal inputs and the utility of ES for benchmarking spatial generalization, even within a geographically compact initial release. For the camera-ready version we will include several quantitative measures of terrain expression (basic statistics of elevation) to help users better understand suitability for generalization to new terrains.
>
> **Broader Applicability:** ES targets classes formed by surface processes that are not unique to the extent of the current dataset. These units are defined by universal geomorphic processes (e.g., hillslope transport, weathering, fluvial deposition) rather than local conditions. For example, colluvium (Qc) is found on hillslopes exceeding material shear strength across the Appalachian Plateau and Carpathians; residuum (Qr) forms in karst systems from the Dinaric Alps to southern China; alluvium (Qal, Qat) is found in river valleys worldwide; and artificial fill (af1) is globally associated with urbanization. Because these are process-based classes, the underlying representations learned from ES are fundamentally transferrable. However, we agree that more tests are needed, which we are actively pursuing.
>
> A central finding of our work is that terrain shape features, including slope (S), elevation percentile (EP), and standard deviation of slope (SDS), generalize more reliably than raw elevation or spectral imagery. These features describe the relative geometry of the Earth’s bare surface, making them inherently less sensitive to local vegetation, lighting, or location-specific conditions. Our experiments confirm that models trained on terrain derivatives consistently outperform those trained on raw imagery (Supp. Tables 4-6, 10-15), supporting their suitability for geographically robust models.
>
> **Uniformity & Expansion:** ES v1.0.1 is currently geographically limited to two counties in Central Kentucky, which reflects the availability of detailed surficial geologic maps and corresponding feature data that experts require for constructing these maps. This uniformity yields an experimental design that ensures labeled classes are coherent across space, and avoids potential discrepancies that may arise when datasets are compiled from multiple regions or mapped by geologists with differences in interpretation. This design also reduces confounding local factors (discussed above) and allows the dataset to more cleanly isolate the effects of modality and fusion strategy. However, this comes with the potential tradeoff that generalization results may overestimate model robustness in other landscapes. We are actively expanding ES to nearly triple the number of patches and introduce more physiographic and surficial complexity (Supp. Fig. 1).
>
> **Naming:** Our intention with "EarthScape" was to reflect the dataset’s focus on Earth surface processes and features, rather than to imply immediate global coverage. For context, naming conventions in ML benchmarks often emphasize the subject rather than geography, including popular datasets like Cityscapes, Places, or SpaceNet. We recognize the potential for misinterpretation and will add geographic qualifiers to dataset filenames (e.g., “earthscape_usa_ky_”) and explicitly define the scope in our documentation for each release.
>
> # "importance of the addressed topic for research or industry"
>
> **Practical Significance:** Surficial geologic materials comprise the thin layer of unconsolidated Earth that interacts directly with human activity. These materials govern critical surface processes related to slope stability (erosion, landslides), hydrology (rainfall runoff, flooding), and environmental sciences (natural resources, industrial contamination). For instance, artificial fill (af1) and colluvial units (Qc, Qca) are key determinants of landslide risk. Alluvial units (Qal, Qat, Qaf) inform floodplain mapping, flood susceptibility, and sediment transport (erosion hazards). Residuum (Qr) and alluvium are also commonly targeted for near-surface mineral exploration (gravel, sand, critical minerals). Government agencies rely on surficial geologic maps for land-use planning, engineering project commissioning, and disaster mitigation and response. Despite their practical relevance, there is a significant lack of these maps around the world. ES addresses this bottleneck by providing the first high-resolution, multimodal benchmark specifically designed to enable automated, scalable surficial geologic mapping using ML.
>
> **United States Federal Initiatives:** ES's relevance is further underscored by its alignment with national-scale geoscientific initiatives. The U.S. Geological Survey’s GeoFramework Initiative aims to produce a seamless nationwide geologic map. Manual mapping is currently too slow to meet this goal at scale, but ES-trained models offer a proof of concept for integrating ML into these workflows. Even in its current form, the dataset enables predictive mapping in unmapped areas that could be refined through expert validation, thereby accelerating production and improving consistency.
>
> **Potential ML Applications:** From a ML standpoint, ES offers a rich and realistic testbed for advancing geospatial AI. Each 256×256 patch includes 38 co-registered channels that capture diverse modalities: 4-band overhead imagery, elevation, terrain derivatives at multiple resolutions, vector-based infrastructure and hydrology, and segmentation masks. The dataset is inherently multilabel, as most patches include multiple geologic units, and it presents severe class imbalance and spatial heterogeneity. All of these characteristics reflect real-world mapping challenges that are not limited to geologic research. ES is also broadly applicable for researchers working on multimodal learning, contrastive representation learning, multi-scale architectures, and domain adaptation. It supports the development and evaluation of fusion strategies that integrate heterogeneous data types with complex spatial structure. It also opens avenues for physically grounded pretraining, where shape-derived terrain features can serve as informative priors for downstream geospatial tasks. These contributions position ES not only as a task-specific resource, but as a general-purpose benchmark for developing and testing robust, interpretable, and scalable vision models.
>
> # "stability in time of the two zones"
>
> ES integrates data acquired between 2010 and 2024, and while such disparities can introduce alignment challenges in rapidly changing environments, their impact is minimal in our case due to the nature of the geologic classes. The surficial geologic maps in ES v1.0.1 were originally created between 2018 to 2024, and the mapped classes represent surficial materials formed by long-term geomorphic processes. Units such as colluvium (Qc, Qca), residuum (Qr), and alluvial deposits (Qal, Qat, Qaf) evolve over timescales of thousands to hundreds of thousands of years. Their spatial boundaries are practically unaffected by changes occurring over the span of a few years. Even where periodic deposition does occur, such as in floodplains, these events result in vertical accretion of new material rather than shifts in unit boundaries (i.e., new flooding deposits Qal on top of older Qal).
>
> Artificial fill (af1) is the exception. It marks locations of recent anthropogenic surface modification and may change more rapidly due to construction and land development. We acknowledge that af1 boundaries do not always capture the most recent development. However, af1 is unlike the other geologic classes in that it reflects zones of elevated geologic uncertainty. Updating af1 to match new construction is relatively straightforward, but reconstructing or interpreting the geology underneath these areas is significantly more challenging and resource-intensive, often requiring subsurface methods such as drilling, coring, or geophysical surveys. Thus af1 serves as a critical class in its own right and offers users a mechanism to flag regions where automated geologic mapping may require additional caution. Our cross-domain Hardin County test set provides some degree of measure on geologic stability, as it was mapped several years before the Warren County area (2018-2022 versus 2022-2024, respectively). Despite the temporal offset, our best multimodal models generalize well to this region (0.059 drop in F1; Table 1). If substantial surficial geologic changes had occurred, we would expect a marked drop in performance. Even af1 maintains cross-generalization (0.009 drop in AUC; Supp. Tables 13 & 14), which suggests that minor changes due to urban expansion do not meaningfully disrupt the alignment between inputs and labels. We will discuss these points in the camera-ready version.

---

> > ### Comment · Reviewer_PsPK · 2025-08-05
> >
> > I thank the reviewers for the rebuttal which clarifies several concerns I had on the stability in time, and to a lesser extend, on the topic importance/application fields and the restricted coverage of the dataset. Regarding the two later, I can see these concerns are also shared by other reviewers and i guess will be discussed at the next stage between reviewers.
> >
> > As a minor note, I found that the author' response was overlong and contained some superficial information that made the reading difficult. I would have prefer short, sharp answers that directly go to the point.

---

> > > ### Author Response · Authors · 2025-08-06
> > >
> > > Thank you for your follow-up. We're glad that many of your initial concerns have been addressed. To respond briefly to the remaining ones:
> > >
> > > ***1. Geographic Scope.*** While ES v1.0.1 is geographically limited (~1,230 km² across 31K patches in two areas), it was deliberately constructed with high sample density, consistent expert labeling, and measurable domain shift.
> > > * ES modalities show strong generalization (0.059 drop in F1 SGMap-Net with terrain shape features vs. 0.187 for SatMAE++ with RGB).
> > >
> > > EarthScape is a *living* benchmark meant to grow:
> > > * ES v1.0.2 (Q4 2025) adds 15 new map areas, tripling coverage (~3,830 km², 93K patches).
> > > * ES v1.0.3 (Q2 2026) will expand beyond the current region (e.g., glaciated terrain in Massachusetts).
> > > * Code and curation tools are open-source and support community contributions.
> > >
> > > ***2. Relevance to ML.*** EarthScape introduces features and challenges relevant to the broader ML community, and not just domain-specific use cases:
> > > * *Multimodal input fusion:* 38 aligned channels per sample support diverse input configurations not available in existing RS datasets.
> > > * *Task diversity:* Supports multilabel classification, semantic segmentation, and regression from the same annotations.
> > > * *Rare class + long-tailed learning:* Multilabel targets with a 60:1 imbalance and a built-in domain-shift split.
> > > * *Shape-centric learning:* Terrain-derived features outperform RGB and DEMs (typical RS modalities), offering new representation learning grounded in shape, not color.
> > >
> > > ***3. NeurIPS D&B Scope.*** EarthScape aligns with the D&B track’s emphasis on thoughtfully designed datasets, open-source tools, and benchmarkable geospatial ML challenges. It offers a reproducible, extensible framework for data-centric AI research grounded in real-world environmental applications.

---

### Comment · Area_Chair_DuD2 · 2025-08-06
**Discussion period has been extended**

Dear reviewers,

The discussion period has been extended. Please read the authors' rebuttal and other reviewers' comments. You are encouraged to provide your further feedback and engage in a discussion with the authors.

For those already engaged in a discussion, thank you for your efforts.

Your AC

---

### Note · Authors · 2025-08-14

We thank the reviewers for their engaged discussion, which clarified key points, improved accessibility, and resulted in score increases and broad recognition of EarthScape’s (ES) value. This note highlights three themes most relevant to the decision and why the work merits acceptance.

**Dataset Value & Difficulty:** ES v1.0.1 provides over 31K patches with 38 co-registered channels per patch, including RGB+NIR, DEM, five terrain derivatives at six spatial scales, hydrography and infrastructure, plus masks, vector labels, and per-class proportions that support classification, segmentation, and regression. This combination of shape-centric features with standard remote sensing (RS) modalities is rare. While coverage is smaller than some CV datasets, ES offers far greater modality depth, presenting a different challenge. SOTA RS models show large cross-domain drops, while our lightweight model significantly narrows that gap, suggesting that performance is far from saturated.

**Broader ML Applicability:** ES enables research in multimodal fusion, domain adaptation, long-tailed learning, and rare-class detection, of broad interest across ML. Its heterogeneous, non-spectral inputs allow benchmarking of architectures that integrate imagery with surface shape signals and vector layers, a gap in current benchmarks. Shape-centric features consistently generalize better than raw imagery, pointing toward physically grounded representation learning relevant to environmental modeling, hazard forecasting, and 3D or geometry-aware tasks.

**Expansion Roadmap & Generalization:** ES v1.0.1 uses two geomorphically distinct regions to test domain shift. v1.0.2 (Q4 2025) will nearly triple coverage. v1.0.3 (Q2 2026) will expand into new physiographic provinces. The open-source curation pipeline enables community contributions for sustained growth. Expansions will broaden coverage and diversity while maintaining stable methodology and evaluation protocols. Publishing now lets the community engage immediately.

ES is a meticulously curated, open, and extensible benchmark with reproducible baselines, addressing an underrepresented yet essential domain. It directly aligns with the D&B track goals of “*New datasets…*”, “*Data-centric AI methods…*”, “*Frameworks for responsible dataset development…*”, and “*Benchmarks on new or existing datasets…*”. The benchmark is fully mature, making this a timely contribution, ready for adoption and poised to serve as a lasting benchmark.

---

### Decision · Program_Chairs · 2025-09-18

**Decision:**

Reject

**Comment:**

This submission introduces EarthScape, a multi-modal, multi-scale dataset for surficial geologic mapping analysis and provides an initial benchmark on that dataset.

It received four thoughtful and very detailed reviews with an average rating of 4.0. The authors have provided a detailed rebuttal to address the questions and concerns raised by the reviewers, which the reviewers acknowledged. I am very impressed by the excellent reviews and high engagement on both sides, the reviewers' and authors' side, during the rebuttal. I very appreciate this and would like to thank everybody involved for her/his engagement. I think the rebuttal was very important for this submission as it has changed the assessment of some of the reviewers.

I have thought a lot about my recommendation for this submission, given the strengths and weaknesses outlined by the reviewers. In particular, Reviewr 3sWo (borderline reject) is concerned about the dataset being small-scale, geographically limited, and missing baseline comparisons. Specifically, the geographic limitation (which was raised as a concern by all other reviewers) might be considered a strong weakness in terms of comparison to other earth observation datasets. On the other hand, for the geological mapping domain, this dataset can be understood as an invitation to the ML community to explore a smaller-scale but rich geological dataset for ML-driven geoscience.

Having said this, and trusting the assessment of the majority of reviewers and their given confidence levels, I recommend this submission to be **accepted**. I ask the authors to include all recommendations given by the reviewers into the final version of the paper.

===== FINAL UPDATE FROM DB Track PCs ====

The final decision for this paper has been taken by the program chairs after consultation with the SACs. All Senior Area Chairs have ranked papers according to the feedback from the AC during the review process. We decided to leave the original meta-review to reflect the opinion of the AC in light of the initial discussions with reviewers and SAC.